# Temperature-Dependent Life Table Parameters of Brown Marmorated Stink Bug, *Halyomorpha halys* (Stål) (Hemiptera: Pentatomidae) in the United States

**DOI:** 10.3390/insects14030248

**Published:** 2023-03-02

**Authors:** Serhan Mermer, Erika A. Maslen, Daniel T. Dalton, Anne L. Nielsen, Ann Rucker, David Lowenstein, Nik Wiman, Mukesh Bhattarai, Alexander Soohoo-Hui, Edwin T. Harris, Ferdinand Pfab, Vaughn M. Walton

**Affiliations:** 1Department of Horticulture, Oregon State University, Corvallis, OR 97331, USAvaughn.walton@oregonstate.edu (V.M.W.); 2Department of Environmental and Molecular Toxicology, Oregon State University, Corvallis, OR 97331, USA; 3Fachhochschule Kärnten, Faculty of Engineering and IT, 9524 Villach, Austria; 4Department of Entomology, Rutgers Agricultural Research and Extension Center, Rutgers University, Bridgeton, NJ 08302, USA; 5Macomb Country Extension Office, Michigan State University, Clinton Township, MI 48036, USA; 6Department of Horticulture, North Willamette Research and Extension Center, Oregon State University, Aurora, OR 97002, USA; 7Department of Ecology, Evolution, and Marine Biology, University of California Santa Barbara, Santa Barbara, CA 93016, USA

**Keywords:** life table, Hemiptera, Pentatomidae, insect ecology, invasive species, phenology

## Abstract

**Simple Summary:**

Brown marmorated stink bug, *Halyomorpha halys* (Stål) (Hemiptera: Pentatomidae), is an invasive pest species that causes significant economic damage to many crops. One of the crucial aspects of management strategies rely on understanding the biology and life-table parameters of *H. halys*. Here, we described the temperature dependent life-table parameters of *H. halys* from New Jersey and Oregon. In addition, comparison with previous data was performed to provide up-to-date information. Altogether, the current data and previous reports were employed to determine developmental thresholds and estimates of optimal temperatures for *H. halys* population development.

**Abstract:**

Brown marmorated stink bug, *Halyomorpha halys* (Stål) (Hemiptera: Pentatomidae), is a generalist pest that causes serious injury to a variety of crops around the world. After the first detection in the USA, *H. halys* became a serious threat to growers resulting in significant crop damage. Understanding the effect of temperature on *H. halys* development will help to achieve successful control by predicting the phenological timing of the pest. Here, life table parameters (survival, development, reproduction, and daily mortality) of *H. halys* were evaluated for New Jersey and Oregon populations in the US. Parameters were determined from field-collected and laboratory-reared individuals. The results indicated that New Jersey populations had higher levels of egg-laying than Oregon populations and exhibited higher and earlier fecundity peaks. Survival levels were similar between populations. Linear and nonlinear fit were used to estimate the minimum (14.3 °C), optimal (27.8 °C), and maximum (35.9 °C) temperatures where development of *H. halys* can take place. An age-specific fecundity peak (M_x_ = 36.63) was recorded at 936 degree days for New Jersey populations, while maximum fecundity (M_x_ = 11.85) occurred at 1145 degree days in Oregon. No oviposition was recorded at the lowest (15 °C) or highest (35 °C) trialed temperatures. Developmental periods increased at temperatures above 30 °C, indicating that such higher temperatures are suboptimal for *H. halys* development. Altogether the most optimal temperatures for population increase (*r_m_*) ranged from 25 to 30 °C. Survival rates of *H. halys* at suboptimal low temperatures of 8 °C (i.e., 61%) is comparable to previous reports. The present paper provides additional data and context from a range of experimental conditions and populations. Such temperature-related *H. halys* life table parameters can be used to provide determine the risk to susceptible crops.

## 1. Introduction

Brown marmorated stink bug, *Halyomorpha halys* (Stål) (Hemiptera: Pentatomidae), is an economic pest of over 300 ornamental and agricultural plants in North America, Asia, and Europe [1,2,3]. *Halyomorpha halys* was first recorded in the United States in 1996, and has become a destructive agricultural and nuisance pest in many states [4]. Second instar nymphs and subsequent life stages of *H. halys* feed on tissues of a wide range of host plants [4,5]. During feeding, *H. halys* deposits saliva within the plant tissues, causing necrosis and scarring [6]. Feeding can cause economic damage that renders the crop unmarketable [6]. In addition, *H. halys* has high reproductive and dispersal capabilities: each female lays an average of 240 eggs per season and adults can fly up to 72 km per day [2,4]. Having such high reproductive rates, mobility, and population reservoirs supported by wild vegetation contribute to the economic impact of *H. halys* on crop production, rapid invasion and population establishment in new regions [7,8,9]. 

The impact of temperature on development, fecundity, survival, and mortality of an invasive pest such as *H. halys* can be used to determine its invasive potential [4] and seasonal risk [9,10]. Previous reports provided similar detailed egg and nymphal developmental periods, survival rates, and degree-day (DD) temperature requirements for *H. halys* [4,10,11,12,13,14]. Eggs and nymphs are unable to complete development to the adult stage at temperatures below 15 °C or above 36 °C [4]. Nymphal developmental periods averaged roughly one week per instar [4]. Adult (non-diapause) life expectancy is approximately 111 days with a pre-oviposition period of 14 days at 25 °C [4,15,16]. 

Life tables are constructed to determine mortality of a population as a response to abiotic or biotic factors [4]. Such data can help determine the number of generations per season, the population structure, abundance, and establishment risk as the pest population spreads [11,12,14,17]. Age-specific fecundity and survival data related to temperature are vital components of seasonal populations and risk models [10,11,12,13,14,18]. The reproductive and developmental parameters of *H. halys* have been studied in temperate climates [4,10,11,12]. However, information about *H. halys* life table parameters at a wider range of temperatures is needed to estimate reproduction, survival, and population dynamics under climatic extremes and in different geographic locations. Extreme temperatures substantially affect survival and reproduction of insects [4,17]. Relating age-specific fecundity and survival data to these extremes are vital for robust population and risk modeling [18]. 

The goal of this study was to provide estimates of minimum, optimum and maximum thresholds for development, reproductive rates, and age-specific survival at temperatures within and below known minimum, optimum and maximum thresholds in the United States where *H. halys* has a prevalent population density. 

## 2. Materials and Methods

### 2.1. Insect Collection and Rearing

Insect collection and rearing of populations from the respective locations were different, precluding direct comparison of data.

#### 2.1.1. *Halyomorpha halys* Oregon Population Rearing

In Oregon, established lab-reared colonies of mixed aged *H. halys* were supplemented with wild insects collected via beat sheet (Model DP1000; BioQuip, Rancho Dominguez, CA, USA) sampling in the Willamette Valley, OR from May to June 2015. Insects were collected from English holly (*Ilex aquifolium* L.), western red cedar (*Thuja plicata* (Donn) D. Don), vine maples (*Acer circinatum* Pursh.), and lilac (*Syringa vulgaris* L.). The Oregon colonies were reared in 30 × 30 × 30 cm bug dorms (#2840R; BugDorm, Taichung, Taiwan) at 22 ± 1 °C and a photoperiod of 16:8 light:dark (L:D). Water was provided in a 473-mL container constructed of plastic (460938; Arrow Plastic Stor, Elk Grove, IL, USA) and topped with a mesh lid. 

A fresh diet of organic Spanish peanut (*Arachis hypogaea* L.), soybean (*Glycine max* (L.) Merr.), pumpkin seed (*Cucurbita* spp. L.), and sunflower (*Helianthus annuus* L.) was provided in an open Petri dish and replaced every 3–4 days. In addition, insects were provided with adequately watered fresh supplies of living jalapeño pepper (*Capsicum annuum* L.), empress tree (*Paulownia tomentosa* (Thunb.) Steud.), organic green bean (*Phaseolus vulgaris* L.), and English holly (*Ilex aquifolium* L.) branch clippings bearing fruit as feeding supplements and oviposition substrates. Intact plants were grown in 800-mL plastic pots filled with Professional Growing Mix Custom Blend soil (Pro Gro, Portland, OR, USA) and were approximately 0.5 m tall. 

Newly emerged adult *H. halys* from field-collected colonies were separated by sex into individual rearing cages. Insects were maintained at 15, 18, 22, 25, 27, 30, and 32 °C, 16:8 L:D and ~85% relative humidity. One- to three-day-old male and female adults (25 of each sex) were placed in new rearing cages within growth chambers (E-30BHO; Percival Scientific, Perry, IA, USA) at each of the seven tested temperatures for the duration of the experiment. Two cages, each containing 50 insects, were placed in each chamber, and exposed to the same temperature conditions. No development was observed at 15 °C in previous study [4], therefore only one cage was tested at this temperature. At 32 °C, one of the two cages contained only 36 insects (18 males and 18 females). Cages were checked daily to determine the reproduction and survival of *H. halys* at these temperatures. Egg masses were removed daily and the number of eggs in each mass was counted. The sex of each dead *H. halys* was determined upon removal. 

#### 2.1.2. *Halyomorpha halys* New Jersey Population Rearing

In New Jersey, a colony maintained and supplemented from yearly field-collected individuals at Rutgers Agricultural Research and Extension Center (Bridgeton, NJ, USA) was reared at 25 °C with 16:8 L:D in temperature cabinets (Precision Scientific, Winchester, VA, USA). Each newly enclosed female was paired with a new male in a 235 mL volume deli cup (Solo TP9R, Lake Forest, IL, USA) with a modified mesh lid. Insects were fed with organic sunflower seeds, a slice of carrot, and green beans. In addition, all diets were replaced three times a week per container. Males of these populations were rotated weekly to assure that male fitness did not impact female fitness. Each temperature regime was initiated starting with 20 mating pairs repeated 23 times during 2012 and 2016. Females were allowed to oviposit until the end of their lifespans. Cages were observed daily, and dead *H. halys* were removed to discourage mold and disease. The number of eggs oviposited, days past eclosion, and female longevity were recorded. Trials were maintained at 17, 20, 22, 27, 30, or 33 °C, 16:8 L:D and ~65% relative humidity. Temperature and light levels within the temperature cabinets were monitored using HOBO data loggers (model UA-002-08; Onset, Bourne, MA, USA).

### 2.2. Survival at Temperatures below Developmental Thresholds

For studies at temperature below developmental thresholds, *H. halys* were collected by hand from building structures in Portland, OR metropolitan area. The timing of insect collections (Mid-November until Mid-December) coincided with the beginning of natural diapause for *H. halys* in this region. *Halyomorpha halys* adults were kept in groups of 20 inside 25 × 50 × 75 mm cardboard shelters fitted with flat 50 × 75 mm cardboard inserts and exposed to 8 °C conditions with 16:8 D:L photoperiod, simulating the average dormant temperatures in Western Oregon. This temperature is known to be below the lower developmental and reproductive thresholds of *H. halys* [4]. Seven shelters were maintained under this rearing regime for three months. To verify that the internal temperature of the shelters was accurate, the probe of a thermocouple data logger (model UA-002-08; Onset, Bourne, MA, USA) was inserted at the base of each shelter and secured with masking tape. We determined mortality of diapausing *H. halys* in all shelters every 3–4 days. Any individual that did not move when its abdomen was prodded was considered dead.

### 2.3. Statistical Analysis

#### 2.3.1. Temperature-Related Daily Survival and Reproduction

Survival and reproductive rates at temperatures within the range suitable for *H. halys* development were recorded. Missing individuals were excluded from statistical analysis. We analyzed survival and fecundity data using nonparametric Kruskal–Wallis ANOVA rank-sum tests to determine differences in survival between temperature treatments. Differences in the means were separated using Tukey’s HSD. All analyses were conducted using Statistica (version Statsoft 7.1; Tulsa, OK, USA). 

The intrinsic rate of population increase (*r_m_*) at each temperature was estimated using mean survival and fecundity values at each temperature. The equation [19,20]
$$r_{m}=\log_{e}R_{o}/T,$$
where *R_o_* is the net reproductive rate and *T* is the mean generation time, was used to determine *r_m_*. Ae The equation $\;R_{o}=\sum l_{x}m_{x}$ was used to determine net reproductive rate. Linear and nonlinear regressions were performed on the present data by using the reciprocal of development time in days (1/T) on temperature. The lower developmental threshold was determined by solving the regression equation for 1/T using the obtained linear regression obtained from the five lowest recorded temperatures from these and previous studies [14,15]. Nonlinear regression was performed on all data from the current and previous *H. halys* studies [14,15] by using the reciprocal of development time in days (1/T) on temperature to determine upper temperature for development. The optimal temperature for development were estimated by fitting the nonlinear estimation model [21] of temperature-related survival of 1/T of 1/T = 0 over temperature.

#### 2.3.2. Physiological Age-Specific Survival and Maternity

The physiological time scale for survival and fecundity was determined by converting the daily heat accumulation from a calendar day basis to DD (single sine) to measure the quantity of accumulated heat units [22] required by *H. halys* to reproduce and develop using thresholds from Section 2.3.1. The age-specific survival (*L_x_*) of nymphs [4] was incorporated with summer and dormant adult survival data and fitted using a two-parameter probability density Gompertz function; F(x I a, b)= exp(−b/a(exp(ax)−1)), where *a* is the shape and *b* is the rate. The data were fitted using R version 3.2.2 using ‘flexsurv’ package [23].

The number of eggs laid per female per day (EFD, age-specific fecundity) was plotted over DD and subsequently fitted using the function $y\;=\;a\;\left(x-\left(b\right)\right)\;c^{1/0.235}$, where x is temperature, and a, b and c are constants. The number of eggs laid per adult female over DD was plotted by summing the eggs laid over 35 consecutive 30-DD periods, starting at 575 DD and ending at 100% mortality at 1615 (Oregon populations) and 1717 (New Jersey populations) DD. The 30-DD periods were used because they allowed adequate resolution and physiological-age comparison. The age-specific fecundity for both groups of insects over time was fitted using the Cauchy distribution, $F(x_{0}\;|\;ϒ)=1/\pi ϒ\left({ϒ}^{2}/\left(x-x_{0}\right)+{ϒ}^{2}\right)$, where *x*_0_ is the location and Υ is the scale. The survival of *H. halys* at temperatures below the lower developmental threshold was fitted to a Gompertz distribution over calendar days. The same two-parameter probability density function was used for survivorship and fecundity at developmentally suitable temperatures.

## 3. Results

### 3.1. Life Span, Daily Mortality, Survival, and Oviposition

*Halyomorpha halys* temperature-related developmental thresholds were 14.26 °C and 35.9 °C, respectively and optimal temperature for development was 27.8 °C. The estimates for linear and nonlinear equations are, respectively, y = 0.0022x − 0.031 (R^2^ = 0.96; F = 112.56; d.f. = 1, 4; *p* = 0.0007) and y = (3.67 × 10^−7^) × x × (x − (15.39)) × ((35.9) − x) × exp (1/(0.298)); R^2^ = 0.92; F = 4.1; d.f. = 1, 15; *p* = 0.0001).

#### 3.1.1. *Halyomorpha halys* Oregon Population

Mean adult survivorship of both males and females decreased with increasing temperature within the developmentally suitable range of 15–32 °C (Table 1, Figure 1A–C). The adult life span ranged from 2 to 187 days for females and 2 to 169 days for males. Adult to mortality at the four highest trialed temperatures (25, 27, 30, and 32 °C) were shorter than at the three trialed lowest temperatures (15, 18, and 22 °C). The mean female adult to mortality at 15, 18, 22, 25, 27, 30, and 32 °C were 89, 71, 60, 40, 41, 28, and 35 days, respectively. Females lived significantly longer at the two lowest temperatures (15 and 18 °C) compared to the two highest temperatures (F_6, 290_ = 23.4, *p* < 0.001, Table 1). Females had significantly shorter survival periods at 22 °C. In addition, female adult to mortality were found shortest at the four highest temperatures of 25, 27, 30, and 32 °C (Table 1, Figure 1A,B). Female longevity did not differ significantly between 25 and 27 °C or between 30 and 32 °C. The mean daily female mortality rates were significantly impacted by temperature, as a proportion of the total number of deaths per day for 15, 18, 22, 25, 27, 30, and 32 °C were 0.164, 0.267, 0.413, 0.676, 0.633, 0.956, and 0.962 females, respectively (χ^2^ = 24.6; d.f. = 1,6; *p* < 0.001; Figure 1A,B). 

The maximum mean fecundity, measured by the number of eggs per female per day (EFD), was highest at 25 °C and decreased at 27, 22, 30, and 18 °C, respectively, with the fewest eggs produced at 32 °C (Figure 2). Fecundity was significantly higher at 25 and 22 °C compared to the other temperatures (F_5, 379_ = 9.6, *p* < 0.001; Table 1). The numerically highest levels of daily oviposition occurred at 25 °C (Figure 2C); however, statistically comparable oviposition levels were also found at 27 and 30 °C, respectively (Figure 2D,E). No oviposition occurred at 15 °C, and the next lowest levels of oviposition occurred at 32 and 18 °C (Figure 2A,F). The longest oviposition period of 113 days was recorded at 22 °C (Figure 2B). These differences were also reflected by the distribution curves of EFD using rank-sum tests (N = 45, d.f. = 1.,5; *p* < 0.001, Figure 2). The highest EFD values were recorded at 25 °C. The lowest EFD values, which also did not differ significantly, were recorded at 32 °C.

#### 3.1.2. *Halyomorpha halys* New Jersey Population

The life span of adults ranged from 4 to 135 days. Female life spans did not differ significantly among 20, 22, and 27 °C treatments, or between 30 and 33 °C. Female adult life span at the longest in two trialed temperatures (20 and 22 °C) and was found shorter at the three trialed temperatures (27, 30, and 33 °C) (F_8, 2326_ = 118.53, *p* < 0.001, Table 1, Figure 1C). The mean female life spans at 20, 22, 27, 30, and 33 °C were 63, 50, 39, 29, and 27 days, respectively. Kruskal–Wallis ANOVA rank-sum tests (χ^2^ = 24.5; d.f. = 1,4; *p* = 0.001; Figure 1C) indicated that mortality rates as a proportion of the total number of deaths per day for 20, 22, 27, 30, and 33 °C were 0.43, 0.44, 0.42, 0.57, and 0.46 females per day, respectively, indicating that temperature significantly impacted mean female adult survival.

In addition, the maximum mean fecundity was highest at 30 °C, followed by 27, 22, 20, and 33 °C, respectively (Figure 3). Fecundity of *H. halys*, as measured by number of eggs per female per day (EFD), was significantly higher at 30 and 27 °C than at other temperatures (F_4, 408_ = 18.69, *p* < 0.001; Table 1). Statistically lower levels of oviposition occurred at 22 and 20 °C (Figure 3A,B). The least oviposition occurred at 33 °C. The longest oviposition period of 126 days was recorded at 20 °C (Figure 3A and Table 1). These differences were also reflected by the distribution curves of EFD using rank-sum tests (N = 72, d.f. = 1,4; *p* < 0.001, Figure 3). The highest EFD values were recorded at 30 °C, and the lowest at 33 °C. Lower and statistically similar EFD values were recorded at 20 and 22 °C. 

### 3.2. Survival, Oviposition, and Fecundity Fitting

Age-specific survival was determined for *H. halys* adults over DD using the Gompertz distribution which consists of the parameters including *a* and *b* (*a* = 0.000814, *b* =−0.0017; χ^2^ = 821; d.f. = 1, 715; *p* = 0.001 Figure 4A) for Oregon populations and New Jersey populations (*a* = 0.000561, *b* =−0.0016; χ^2^ = 1272; d.f. = 1, 1177; *p* = 0.001 Figure 4B). The number of DD needed to develop from egg to the first nymphal stage was 54 from fitted data. For New Jersey, the number of DD to develop from egg-laying to the second, third, fourth, and fifth instars, and then to adults were 107, 216, 303, 394, and 520 DD, respectively. The number of DD from egg to full mortality for female *H. halys* was 1615 in Oregon and 1717 DD in New Jersey.

The regression of oviposition in the Oregon population is described by the functions y = 0.00000156 (6.513) (42.0577) − x)^(1/0.253)^ (R^2^ = 0.92, F_1, 31_ = 9.7, *p* = 0.002) and y = 1/(3.14 × (4.61591) × (1 + ((x − (9.30887))/(4.61591))^2^))(538.946) (R^2^ = 0.79, F_1, 35_ = 84.78, *p* = 0.001; Table 2, Figure 5)) in the New Jersey population. Estimated temperature-specific oviposition was 11.41 EFD at 15 °C, increasing to 52.44 EFD at 18 °C, 95.14 EFD at 22 °C, and peaking at 107.23 EFD at 25 °C. At 27 °C, EFD decreased to 101.11 EFD, 64.41 EFD at 30 °C and 17.95 EFD at 32 °C.

The gross fecundity for Oregon *H. halys* populations on a DD scale, as described by the Cauchy distribution and the parameters were found as *l* = 16.2; *s* = 2.3; χ^2^ = 9.705; d.f. = 1, 31; *p* = 0.97 (Figure 5A). In addition, parameters for New Jersey were found as *l* = 9.3; *s* = 4.6; χ^2^ = 84.4; d.f. = 1, 35; *p* = 0.001 (Figure 5B). The first recorded age-specific fecundity (*M_x_*; Figure 5A) of *H. halys* in Oregon occurred at 646 DD and was 0.27 EFD. Fecundity levels increased to an estimated maximum of 11.85 EFD at 1145 DD. *M_x_* decreased to 0 EFD at 1595 DD. Age-specific fecundity (*M_x_*; Figure 5B) of *H. halys* in New Jersey at 646 DD was 12.96 (EFD) and increased to a maximum of 36.63 EFD at 936 DD. *M_x_* decreased to 0.72 EFD at 1717 DD.

The DD winter survival curve provided a good fit for Oregon populations of *H. halys* using the Gompertz distribution and has the parameters (*a* = 0.0297, *b* =−0.005; χ^2^ = 58.3; d.f. = 1, 29; *p* = 0.001; Figure 6). There was 50, 80, and 95% mortality at 54, 82, and 105 days, respectively.

The net reproductive rate (*R_o_*), mean generation time (*T*), and intrinsic rate of population increase (*r_m_*), for *H. halys* differed between populations (Table 3). New Jersey populations laid more eggs than Oregon populations. For both populations, increased temperatures resulted in higher *R_o_* values up to 27 °C, but the reproductive rate decreased above that temperature. The mean generation time, *T,* decreased with increasing temperatures, although there was a slight increase of mean generation time at 33 °C for the New Jersey population. An increase in temperature resulted in an increase of *r_m_* values among Oregon populations, up to 30 °C: the highest *r_m_* values were recorded at 27 and 30 °C. Similarly, the *r_m_* value was estimated at 30 °C for New Jersey population, with *r_m_* decreasing at 33 °C.

## 4. Discussion

The current study examines the survival and reproduction of adult *H. halys* from two different regions in the United States, Oregon and New Jersey, under a range of controlled and constant temperatures. The datasets from the present and previous reports were generated using different rearing techniques, precluding direct comparison. Novel data from the combined studies offer insights regarding respective rearing techniques on life table parameters at comparative temperatures. In addition, immature data fitted from the present study estimates the lower, optimum, and upper thresholds for development which were supported by previous reports [4,14,15]. Additionally, temperature-dependent immature survival and development, adult survival under typical winter conditions, and reproduction were evaluated. Altogether, the current and other datasets from previous reports present a more detailed picture of the prospective impacts of key abiotic factors on *H. halys* life table parameters [4,10,11,12]. 

Adult longevity of *H. halys* from the current study falls within the lower range when compared to other studies [4,14,15]. Such differences might depend on factors including diet, adaptation of species to certain climatic conditions, and genetic background [24,25,26,27]. The data from the current study are however comparatively similar and within realistic developmental parameters. Altogether, *H. halys* developmental periods in all currently known studies are from 42.3 to 30.9 days at 25 and 30 °C, respectively [4,10,11,12]. The pre-ovipositional period of *H. halys* from the current and two other studies are similar at 25 °C [4,10]. Comparison of the ovipositional period of *H. halys* in the current study with other studies differ by about 19 days but are not directly comparable [12]. In the current study, the mean *H. halys* fecundity at 25 °C was found 48.6 eggs in Oregon and 212.3 eggs in New Jersey, compared to 212.3 eggs [4], 32.3 eggs [11], 308.6 eggs (23 °C) and 278.9 eggs (27 °C) [12]. 

Adult survival rates were found similar for both Oregon and New Jersey populations. Relatively short survival periods and lower reproductive rates were recorded at the lowest (15 °C) and highest (33 °C) trialed temperatures. Low survival and slower developmental rates were recorded close to the trialed thermal thresholds indicating that such temperatures are close to the developmental limits for *H. halys*, and are comparable to previous studies [4,11,12]. Although datasets are not directly comparable, immature insects reared in New Jersey were likely received a relatively more optimal diets, compared to Oregon insects, which were collected from the field. Likely such differing methodologies resulted in higher fecundity and increased longevity levels for New Jersey insects. 

The respective physiological age-related maternity curves indicate differing levels of reproductive potential for the two studied populations of *H. halys*, but both fall within the range of realistic levels when compared to previous studies [4,11,12]. The winter survival data collected from Oregon offers valuable insights into the ability of *H. halys* to survive at temperatures (typical of milder winter conditions), which were observed in the Willamette Valley of Oregon [28]. Survival data from the current study indicated the survival rate as 50 days [29]. However, there are substantial differences in study design that preclude direct comparison with previous studies [29]. Age-and temperature-related data from both populations indicate which temperatures are most suitable for adult survival and oviposition. The *r_m_* values, indicative of optimal temperatures for population increase is between 25 and 30 °C for both *H. halys* populations.

Other factors including the genetic make-up, rearing conditions [24], field collections, and microorganism infection (e.g., the presence of *Nosema maddoxi*) [30] status of each population can significantly impact influence fecundity and preclude direct comparison with the previous reports [24,25,26,30,31]. The rearing conditions in New Jersey and Oregon were different in terms of methodology (group vs. individual pairs) and temperature, e.g., 22 °C in Oregon and 25 °C in New Jersey. This may have resulted in some level of preconditioning of insects in each of the respective studies. The data presented here should therefore be considered with such differences in mind. Oregon populations were collected from the field as in all life stages, furthermore, immature life stages were reared to the adult stage, and field-collected adults were allowed to reproduce. The New Jersey populations were reared under optimal laboratory conditions throughout. The underlying reason for these differences, however; falls outside the scope of this study. Additional research is essential to determine if the differences found in the fecundity between Oregon and New Jersey populations resulted from methodological, genetic, or other differences between locations and populations. 

The interaction of cyclic temperatures with nonlinear characteristics of development and reproductive curves of ectothermic insects may introduce significant deviations from the results obtained in such studies. Studies across a broader set of fluctuating temperature regimes are therefore encouraged so that a more realistic effect of temperature on the biological parameters of *H. halys* could be understood. Earlier research on biocontrol agents provides compelling evidence that daily temperature fluctuations significantly affected the development times and longevity of insects, resulting in marked deviations when comparing with the constant temperature regimen counterparts [32,33]. As previously reported, evaluating and modeling the effects of realistic fluctuating temperatures may improve estimations of the establishment, spread, and impact of insects across different regions [34].

Ambient temperatures are only estimates of the temperatures experienced by this insect. Adult *H. halys* is highly mobile and could migrate to more optimal temperatures, impacting final outcomes. Sunning on leaves is a behavior that *H. halys* uses to warm itself with solar radiant heating, and, similarly, the insect may also utilize conductive heating by perching on man-made and natural surfaces with thermal mass [35,36]. Similarly, the insect can manipulate its position in the tree canopy to take advantage of the cooling effects from leaf transpiration and shade [37]. This aspect is however outside the scope of the current study.

## 5. Conclusions

The present paper offers new data and comparative analysis on *H. halys* from multiple regions worldwide, building upon previous research conducted on the species. *Halyomorpha halys* is an economically crucial insect species that causes serious damage to multiple crops in the United States [38,39]. It has been stressed that having sufficient life-table parameters are necessary to develop more accurate prediction models for invasive pest species in the case of further climatic change scenario [40,41]. These comparative and unique parameters of *H. halys* can be used to explore possible scenarios at a range of temperature extremes and help researchers in the future to strategize their pest control methods. Population distribution and seasonal estimation models have been developed [9,27] for *H. halys* but can likely be elaborated and refined with the addition of the parameters from these studies. 

## Figures and Tables

**Figure 1 insects-14-00248-f001:**
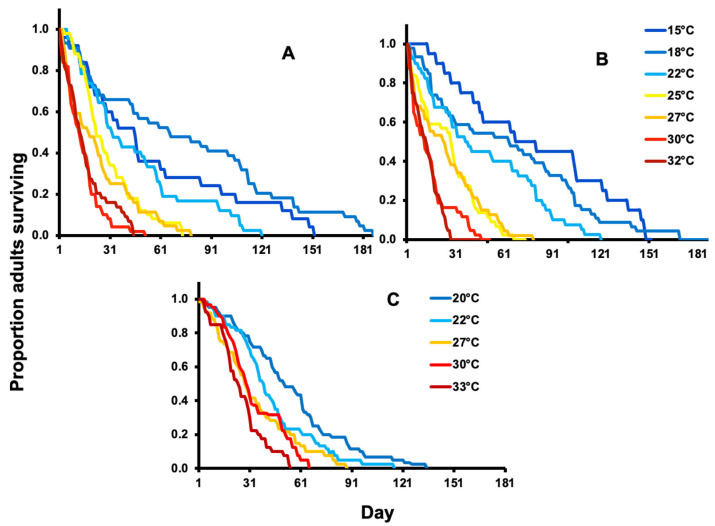
Proportion of *H. halys* adult survival at constant temperatures (°C) of Oregon female (**A**) and male (**B**) and New Jersey female (**C**).

**Figure 2 insects-14-00248-f002:**
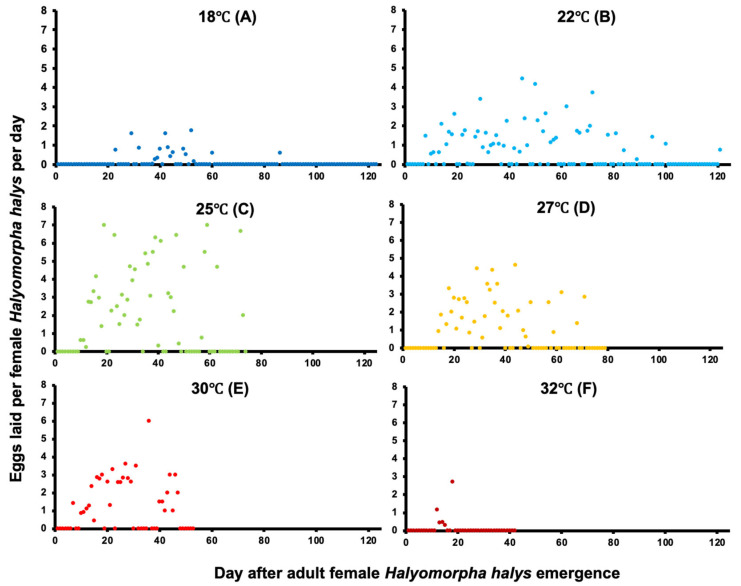
Mean eggs per female per day (EFD) of *H. halys* in Oregon at six constant temperatures: 18 °C (**A**; dark blue), 22 °C (**B**; light blue), 25 °C (**C**; green), 27 °C (**D**; yellow), 30 °C (**E**; red) and 32 °C (**F**; dark red).

**Figure 3 insects-14-00248-f003:**
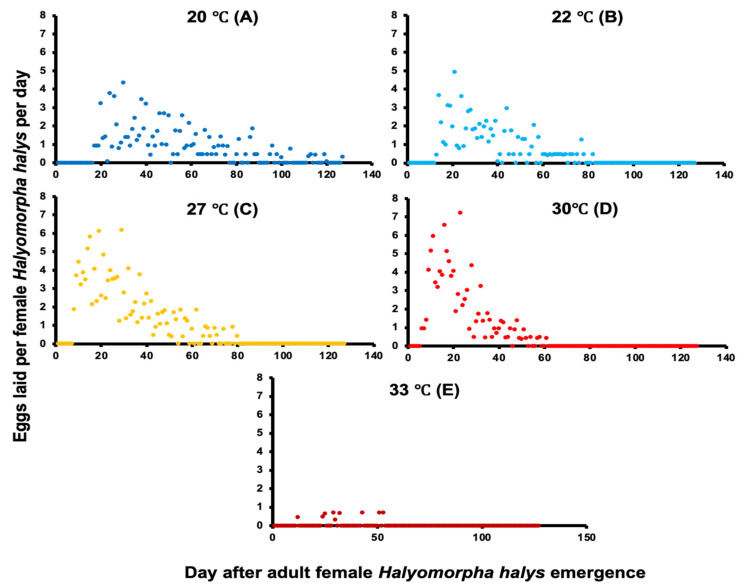
Mean eggs per female per day (EFD) of *H. halys* in New Jersey at five constant temperatures: 20 °C (**A;** dark blue), 22 °C (**B;** light blue), 27 °C (**C;** yellow), 30 °C (**D;** red), and 33 °C (**E;** dark red).

**Figure 4 insects-14-00248-f004:**
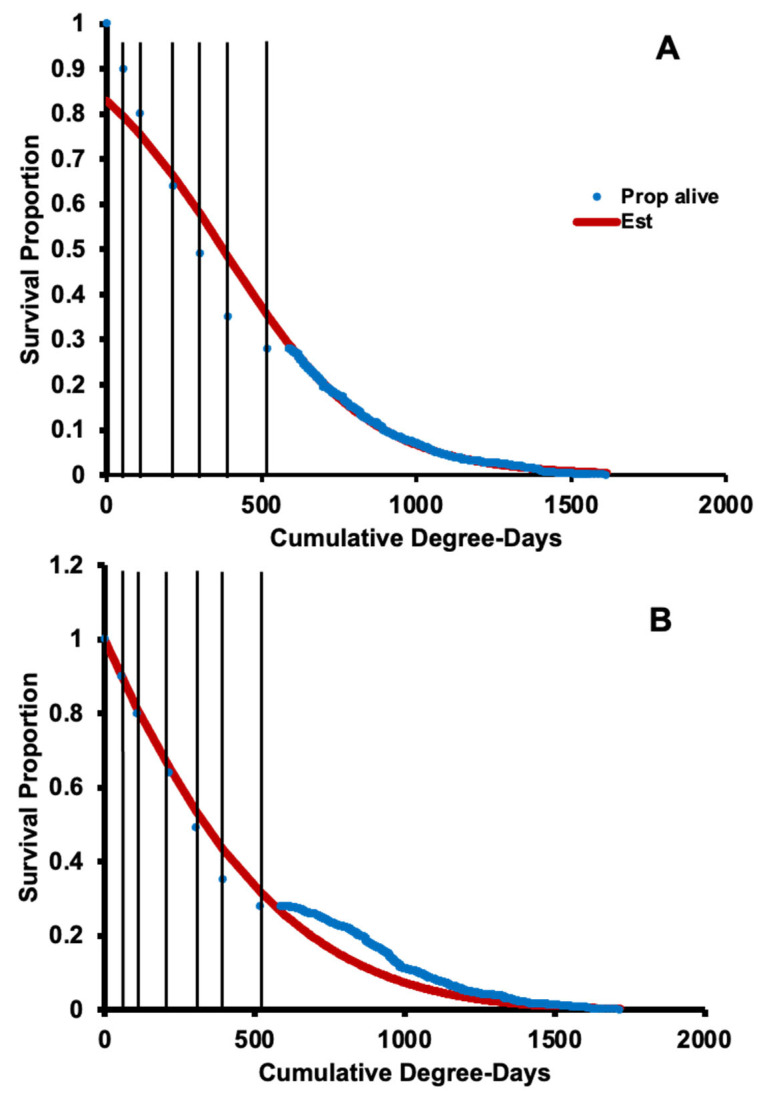
Age-specific survival (*L_x_*, blue dots) and Gompertz distribution fit (red line) of Oregon (**A**) and New Jersey (**B**) populations of *H. halys* over physiological time (degree days). Dots indicate the survival proportion, and curved (fitted) line represents estimated values. Vertical lines indicate life stages from left to right (New Jersey data): first through fifth instar nymphs and adults, until 100% recorded mortality (right line).

**Figure 5 insects-14-00248-f005:**
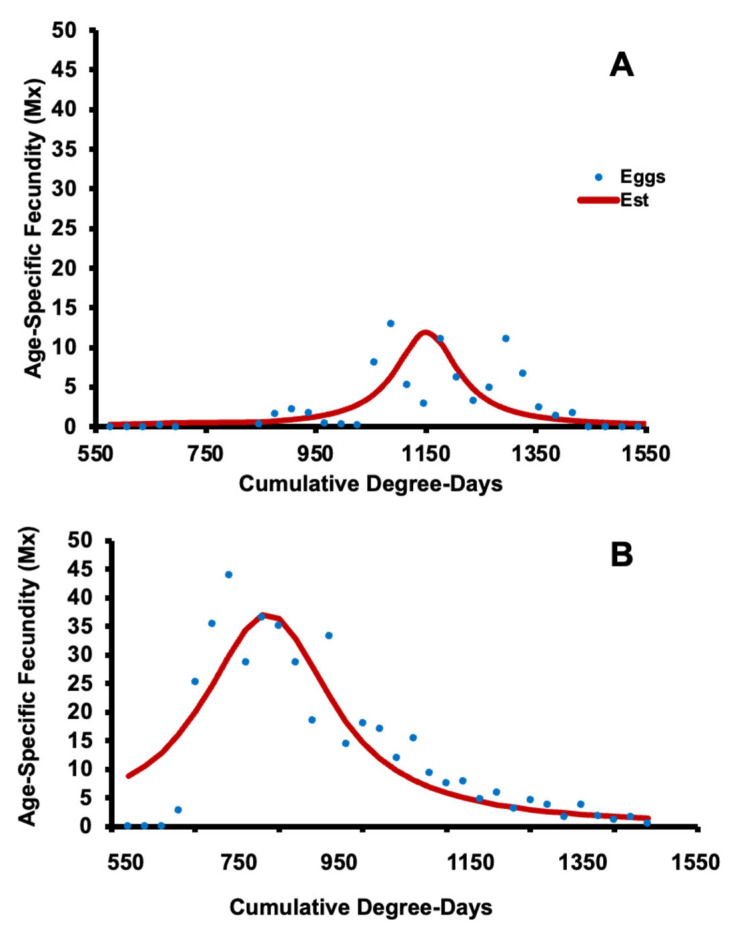
Age-specific fecundity (*M_x_*, blue dots) and Cauchy distribution fit (red line) of Oregon (**A**) and New Jersey (**B**) populations of *H. halys* over physiological time (degree days).

**Figure 6 insects-14-00248-f006:**
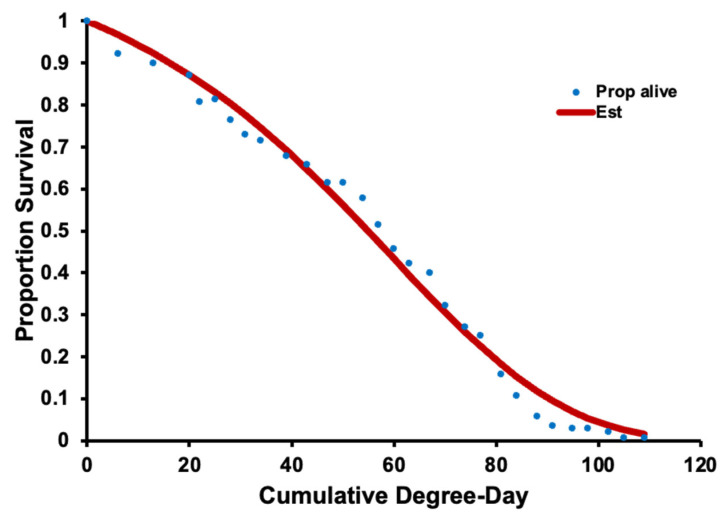
Time-specific survival (*L_x_*, blue dots) and Gompertz distribution fit (red line) of adult *H. halys* in Oregon at 8 °C and 16:8 D:L light regimes.

**Table 1 insects-14-00248-t001:** *Halyomorpha halys* adult female survival and reproductive parameters (±SD) at constant temperatures from Oregon and New Jersey (°C) (EFD = mean eggs laid per female per day; EF = mean eggs per female; N = number of individuals).

Temperature (°C)	Egg to Adult (Days)	Adult to Mortality (Days ± SD)	EFD (Mean Eggs Laid per Female per Day)	EF (Mean Eggs per Female)	N
Oregon					
15	-	55.5 ± 47.0	-	-	25
18	-	75.1 ± 57.3	0.10	7.8	44
22	-	43.8 ± 32.9	0.77	33.9	42
25	-	28.3 ± 17.6	1.7	48.6	50
27	-	22.8 ± 20.5	0.93	21.1	44
30	-	13.8 ± 10.7	0.87	11.9	50
32	-	15.7 ± 12.8	0.13	2.1	44
New Jersey					
17	121.5 ± 0.5 ¢	-	-	-	-
20	81.2 ± 0.8 ¢	52.7 ± 28.4	1.0	54.4	61
22	-	42.2 ± 23.4	1.1	48.5	60
25	44.9 ± 0.8 ¢	47.97	-	212.3	28
27	35.8 ± 0.5 ¢	32.8 ± 21.3	2.2	72.3	59
30	33.4 ± 0.5 ¢	31.1 ± 15.5	2.4	73.6	60
33	37.8 ± 0.9 ¢	24.9 ± 13.6	0.04	1.1	7

¢ [4].

**Table 2 insects-14-00248-t002:** Model fit of oviposition in two population (Oregon and New Jersey, USA).

Population	Model Fit	R^2^	*p*
Oregon	y = 0.00000156 (6.513) (42.0577) − x)^(1/0.253)^	0.92	0.002
New Jersey	y = 1/(3.14 × (4.61591) × (1 + ((x − (9.30887))/(4.61591))^2^)) (538.946)	0.79	0.001

**Table 3 insects-14-00248-t003:** Estimated life table parameters in relation to temperature (°C) of *Halyomorpha halys* (*R_0_* = net reproductive rate; *T* = mean generation times in days; and *r_m_* = intrinsic rate of population increase) from four comparable studies.

Temperature (°C)	*R_o_*	*T*	*r_m_*
18	7.6 _†_	142 _†_	0.02 _†_
20	68.8 _§_, 54.4 _¥_	145 _§_, 140 _¥_	0.028 _§_, 0.028 _¥_
22	48.5 _¥_, 33 _†_	111 _¥_, 123.1_†_	0.06 _¥_, 0.04 _†_
23	128 _§_	79 _§_	0.06 _§_
25	49 _†_, 60 _¢_	103.7 _†_, 60 _¢_	0.065 _†_, 0.07 _¢_
27	151 _§_, 72.3 _¥_, 21.1 _†_	56.8 _§_, 69 _¥_, 72.1_†_	0.087 _§_, 0.063 _¥_, 0.074 _†_
30	58.12 _§_, 74.6 _¥_, 11_†_	50.94 _§_, 33.1 _¥_, 37.7_†_	0.079 _§_, 0.053 _¥_, 0.02 _†_
32	2 _†_	48.5 _†_	0.001 _†_
33	6.8 _§_, 2.05 _¥_	51.2 _§_, 44 _¥_	0.0339 _§_, 0.016 _¥_

_§_ [12]. _¥_ This study, New Jersey. _†_ This Study, Oregon. _¢_ [4].

## Data Availability

The data presented in the current study are available upon request from the corresponding author.

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
