# Peer review of "Temperature-Dependent Life Table Parameters of Brown Marmorated Stink Bug, Halyomorpha halys (Stål) (Hemiptera: Pentatomidae) in the United States"

_insects, 2023, doi:10.3390/insects14030248_

Round 1
Reviewer 1 Report (Previous Reviewer 1)
Authors have done a nice job addressing all of my original comments. I have no further suggestions to improve the paper. Thank you.
Author Response
The authors would like to thank reviewer#1 for substantially contributing the quality of the manuscript.
Reviewer 2 Report (New Reviewer)
The manuscript titled "Temperature-dependent life table parameters of brown marmorated stink bug, Halyomorpha halys (Hemiptera: Pentatomidae) 3 Stål in the United States" (insects-2165612) is an audacious attempt to broaden our knowledge of the life history parameters of H. halys populations in the US. The quality of the presentation is great, however, the estimate of developmental parameters (egg to adult) from the linear/ non-linear models falls outside the biologically relevant range for the species based on data from the previous or current study. Other parameters estimated based on those lower or upper development thresholds will also need to be estimated again, the results updated and the discussion revisited.
Major and minor comments are provided below:
Line 82. period missing at the end of the line.
Line 150. for
Lines 196-198: The statement is not complete. Fix it.
Also, the estimates of lower and upper-temperature thresholds for the egg to adult development do not make any biological sense. The eggs would neither develop at 10.5 C nor the early or late instars (reared even at optimal temperatures and moved to 40.2) would survive to adults at 40.2C. Indeed, temperatures near or above 36C cause nearly 100% mortality. It is recommended to fit new models and provide meaningful estimates. Neither the data from previous studies nor the current study supports these estimates of lower and upper threshold temperatures for development.
217-22: Table 1.
The study pertains to the US population of H. halys and Table 1 title suggests data exclusive to New Jersey and Orgeon populations. The inclusion of data for the Swiss population of H. halys seems inappropriate in this table 1 and suggests removal. Those estimates appear worth reference for comparison in the discussion.
267: Reestimating the thresholds would require refitting this model to derive new estimates.
279: a straight line or fitted curve?
Author Response
We would like to thank reviewer#2 for the comments and feedback. Please see the responses below.
The manuscript titled "Temperature-dependent life table parameters of brown marmorated stink bug, Halyomorpha halys (Hemiptera: Pentatomidae) 3 Stål in the United States" (insects-2165612) is an audacious attempt to broaden our knowledge of the life history parameters of H. halys populations in the US. The quality of the presentation is great, however, the estimate of developmental parameters (egg to adult) from the linear/ non-linear models falls outside the biologically relevant range for the species based on data from the previous or current study. Other parameters estimated based on those lower or upper development thresholds will also need to be estimated again, the results updated and the discussion revisited.
Major and minor comments are provided below:
Line 82. period missing at the end of the line.
R: Thanks for pointing that out. The period was added at the end of the sentence. Please see LN 84.
Line 150. For
R: “for” added to the sentence. Please see LN 153.
Lines 196-198: The statement is not complete. Fix it.
R: The authors thank the reviewer for the comment, the sentence was reworded as suggested. LN 195-199.
Also, the estimates of lower and upper-temperature thresholds for egg to adult development do not make any biological sense. The eggs would neither develop at 10.5 C nor the early or late instars (reared even at optimal temperatures and moved to 40.2) would survive to adults at 40.2C. Indeed, temperatures near or above 36C cause nearly 100% mortality. It is recommended to fit new models and provide meaningful estimates. Neither the data from previous studies nor the current study supports these estimates of lower and upper threshold temperatures for development.
R: The authors would like to thank the reviewer for the comment on the lower and upper-temperature threshold of our analysis in the previous paper. In the initial estimation, the authors did not include 0 data points at 12 and 35 C. In the subsequent analysis, we included 0 values, and this resulted in the new parameters which are included in the new version of the manuscript. Please see LN 37-38, 195-199.
217-22: Table 1.
The study pertains to the US population of H. halys and Table 1 title suggests data exclusive to New Jersey and Oregon populations. The inclusion of data for the Swiss population of H. halys seems inappropriate in this table 1 and suggests removal. Those estimates appear worth reference for comparison in the discussion.
R: Swiss population results were removed from Table 1. Please see Table 1.
267: Reestimating the thresholds would require refitting this model to derive new estimates.
R: The sentence was reworded for clarity as no threshold was determined in this section except age-specific fecundity was determined. Please see LN. 266-269.
279: a straight line or fitted curve?
R: The wording was changed as suggested. Please see Fig. 4 caption, LN 278
Reviewer 3 Report (New Reviewer)
This is a very interesting and well written paper on the development of brown marmorated stink bug (BMSB), Halyomorpha halys, at different temperatures. I don’t have any major comments. Below are a few very minor comments.
Line 119: How long has the New Jersey lab colony being maintained? This could influence their oviposition potential. It is evident in Table 1. Eggs per female are notably high for lab-reared New Jersey insects compared with field-collected Oregon insects. This ties into preferences of field-collected BMSB that appear to be more selective in oviposition, whereas those from the lab may have potentially lost their selective ability, due to prolonged lab rearing, as they attempt to maximize their progeny success by laying more eggs. It would be interesting to see how host preferences and performances of lab-reared BMSB differ from field-collected BMSB. That is, whether selective oviposition behavior of field BSMB females results in optimal progeny performance.
Lines 126-127, Line 150, Line 161: Check the text.
Provide equations for regression models (line 161) in the Methods section.
Lines 194 – 195: Please present your results only.
Author Response
We would like to thank reviewer#3 for the comments and feedback. Please see the responses below.
This is a very interesting and well-written paper on the development of brown marmorated stink bug (BMSB), Halyomorpha halys, at different temperatures. I don’t have any major comments. Below are a few very minor comments.
Line 119: How long has the New Jersey lab colony being maintained? This could influence their oviposition potential. It is evident in Table 1. Eggs per female are notably high for lab-reared New Jersey insects compared with field-collected Oregon insects. This ties into preferences of field-collected BMSB that appear to be more selective in oviposition, whereas those from the lab may have potentially lost their selective ability, due to prolonged lab rearing, as they attempt to maximize their progeny success by laying more eggs. It would be interesting to see how host preferences and performances of lab-reared BMSB differ from field-collected BMSB. That is, whether the selective oviposition behavior of field BSMB females results in optimal progeny performance.
R: The authors appreciate the observation from the reviewer; however, we do know that New Jersey populations were supplemented by field collections. The details are added in the edited manuscript. It would be interesting to see how host preference would change when insects are reared on certain diets, but this question falls outside of the scope of this paper. Please see LN 119-133.
Lines 126-127, Line 150, Line 161: Check the text.
R: Sentences were reworded. Please see lines 128-129; 153-155; 163-165.
Round 2
Reviewer 2 Report (New Reviewer)
Dear Authors,
The authors are appreciated for their quick response time on the suggested edits. They have re-estimated the parameters of concern and produced biologically realistic estimates of lower and upper thresholds for egg-adult development of H. halys.
A last but important edit to suggest is:
Line 175-176: What was the DD estimation approach used? Average or Single Sine or any other? Suggest incorporating to add clarity.
Best Regards,
Reviewer
Author Response
The authors appreciate the additional comment from reviewer#2. We used single-sine DD estimation and the required detail was added to the manuscript. Please see LN 175.
This manuscript is a resubmission of an earlier submission. The following is a list of the peer review reports and author responses from that submission.
Round 1
Reviewer 1 Report
The authors have graphed and presented their results clearly, drawing some attention to the implications of their findings. I found the study of interest and a good contribution to the knowledge of bioecology of BMSB. The methods used are appropriate for the objectives of the work and, in general, well depicted. The resulting figures are sufficient, informative, and of good quality helping to follow the reasoning throughout the manuscript. The discussion of results and comments on future research should be improved if the paper is to be accepted for publication in Insects.
My only concern is that the authors are extrapolating the applicability of their results beyond what the design supports. These are only data from a set of highly artificial constant laboratory conditions, so the inference power of the paper is very limited, but authors do not acknowledge this detail at all and need to be more forthcoming. The effects of fluctuating temperature profiles on BMSB development, longevity, and reproductive capacity were not investigated in this study. This is a critical limitation of the study, and the authors must concede and discuss this. The interaction of cyclic temperatures with nonlinear characteristics of development and reproductive curves of ectothermic insects can introduce significant deviations from the results obtained here, especially at the lower and higher temperatures of development and life activity functions. Studies across a broader set of fluctuating temperature regimes are therefore encouraged so that more realistic effect of temperature on biological parameters of BMSB could be understood, as this is the closest to the daily temperature fluctuations that occur in the field. So, I am suggesting to the authors to tone-down the language a little and admit that there are still substantive uncertainties to be considered.
Some of the authors statements would be much stronger if they tie their work to the body of literature that has built up on the bioecology and reproductive biology of other mass-produced biocontrol agents (BCAs) for field releases in California. Some examples are J. Econ. Entomol. 112: 1560-1574 (mass produced ectoparasite BCAs) or J. Econ. Entomol. 112:1062-1072 (mass produced endoparasite BCAs), but there are others too. These studies provide strong evidence that daily temperature fluctuations significantly affected development times and longevity of insect BCAs, resulting in marked deviations when compared constant temperature regimen counterparts. Consequently, phenology predictions using models fit to constant temperature data could erroneously predict generation times for BCAs during the season. Consequently, evaluating and modeling the effects of realistic fluctuating temperatures scenarios on natural enemy and pest developmental and reproductive biology may improve the predictions of establishment, spread, and impact of BCAs (but also pests) across different regions. This article should provide details on all these fronts to provide the proper context for the work. This is not to diminish the data gathered in this study, they are of value. But it is important for the authors not to overgeneralize, and to warn the reader, including regulatory agencies, against doing so as well. Adding these details will improve the discussion.
Good luck!
Author Response
Reviewer#1
The authors have graphed and presented their results clearly, drawing some attention to the implications of their findings. I found the study of interest and a good contribution to the knowledge of bioecology of BMSB. The methods used are appropriate for the objectives of the work and, in general, well depicted. The resulting figures are sufficient, informative, and of good quality helping to follow the reasoning throughout the manuscript. The discussion of results and comments on future research should be improved if the paper is to be accepted for publication in Insects.
R: The authors acknowledge the comment of the reviewer#1 and manuscript sections were edited and necessary changed were added to the sections.
My only concern is that the authors are extrapolating the applicability of their results beyond what the design supports. These are only data from a set of highly artificial constant laboratory conditions, so the inference power of the paper is very limited, but authors do not acknowledge this detail at all and need to be more forthcoming. The effects of fluctuating temperature profiles on BMSB development, longevity, and reproductive capacity were not investigated in this study. This is a critical limitation of the study, and the authors must concede and discuss this. The interaction of cyclic temperatures with nonlinear characteristics of development and reproductive curves of ectothermic insects can introduce significant deviations from the results obtained here, especially at the lower and higher temperatures of development and life activity functions. Studies across a broader set of fluctuating temperature regimes are therefore encouraged so that more realistic effect of temperature on biological parameters of BMSB could be understood, as this is the closest to the daily temperature fluctuations that occur in the field. So, I am suggesting to the authors to tone-down the language a little and admit that there are still substantive uncertainties to be considered.
Some of the authors statements would be much stronger if they tie their work to the body of literature that has built up on the bioecology and reproductive biology of other mass-produced biocontrol agents (BCAs) for field releases in California. Some examples are J. Econ. Entomol. 112: 1560-1574 (mass produced ectoparasite BCAs) or J. Econ. Entomol. 112:1062-1072 (mass produced endoparasite BCAs), but there are others too. These studies provide strong evidence that daily temperature fluctuations significantly affected development times and longevity of insect BCAs, resulting in marked deviations when compared constant temperature regimen counterparts. Consequently, phenology predictions using models fit to constant temperature data could erroneously predict generation times for BCAs during the season. Consequently, evaluating and modeling the effects of realistic fluctuating temperatures scenarios on natural enemy and pest developmental and reproductive biology may improve the predictions of establishment, spread, and impact of BCAs (but also pests) across different regions. This article should provide details on all these fronts to provide the proper context for the work. This is not to diminish the data gathered in this study, they are of value. But it is important for the authors not to overgeneralize, and to warn the reader, including regulatory agencies, against doing so as well. Adding these details will improve the discussion.
R: The authors appreciate the input that reviewer provided and the authors agreed that adding such details would strengthen and improve the discussion immensely. Please see the LN. 414-428.
Good luck!

Reviewer 2 Report
The manuscript reports adult Halyomorpha halys survival and reproduction from two geographical regions (New Jersey and Oregon state) of the U.S. under a range of controlled temperatures. The authors describe the effect of temperature on survival and fecundity of two populations.
Major
1. The authors follow the guideline of the Journal Insects (format, reference and etc).
2. Statistical analysis of life table parameters is need to modify.
3. The authors explain the results clearly.
4. The authors mixed the contents of results and discussions. Please separate them.
5. The resolution and performance of figures are not clear.
6. The authors give more discussion related to the results of this study.
7. There is a logical contradiction in the discussion.
Minor
Line 6: Ferdinand Pfab7 à Ferdinand Pfab6.
Please add “Simple Summary”.
Line 41: Halyomorpha halys à H. halys.
Line 59-62: Please give references.
Line 62-63: This sentence is redundant.
Line 138: There are no results of development under extreme temperatures.
Line 163: Please check other equation.
Line 166-167: There is no results of the nonlinear estimation model to the intrinsic rate of population increase.
Line 174-175: The authors explain why they choose Gompertz function.
Line 178: What is “0.235” in the function?
Line 184: The authors explain why they choose Cauchy function.
Line 196-197: How the authors define the definition of highest and lowest temperatures in here?
Line 197-198: “Adult life span” and “adult to mortality” (in table 1) are the same concept or not?
Table 1: Why the authors add “Egg to adult” data?
Table 1: The column of EFD in New Jersey at 17°C is 100. Is it correct?
Line 200-202: This sentence is redundant.
Line 218: at 25 and 30 à at 25 and 22?
Line 219-221: This sentence is redundant.
Line 223: Would you check the contents of table 1?
Figure 2. Color dots are better to read the data.
Line 234: How the authors define the definition of highest and lowest temperatures in here?
Line 248: The highest EFD values were recorded at 27 and 30 à The highest EFD value was recorded at 30.
Figure 3. Color dots are better to read the data.
Line 259-261: Please add the results of New Jersey population.
Figure 4: Color dots and lines are better to read the data.
Figure 4B. There are two nonlinear curves. Which one is correct?
Line 268-273: Please make a table for readers.
Line 269: F à F, p à P.
Line 270: R2 à R2, F à F, p à P.
Line 276-277: What are the parameters (l and s)?
Line 279-282: The contents are estimated EFD?
Figure 5: Color dots and lines are better to read the data.
Figure 6: Color dots and lines are better to read the data.
Line 300-302: This sentence does not relate to the results of this study.
Line 303-304: Which population did you compare with other pentatomids?
Line 307-308: The authors discuss about this sentence.
Line 310-321: The authors give discussion focusing on the results of this study.
Line 322-329: Please give quantitative explanation the difference or similarity between two populations.
Line 331-344: Why the authors add this part?
Line 345-366: The authors explain this part clearly and logically.
Line 367-377: This part move to the results.
Table 2: Please reconstruct the table 2 for readers.
Line 368-370: Please rewrite this sentence.
Line 375: “32” à “30”?
Line 389-391: Please give some information about these part in the results.
Line 397-399: There is no contents of this part in the results and discussion.
There is no conclusion part.
References: The authors follow the guideline of “Journal Insects”.
Author Response
Reviewer#2
Major
- The authors follow the guideline of the Journal Insects (format, reference and etc).
- Statistical analysis of life table parameters is needed to modify.
- The authors explain the results clearly.
- The authors mixed the contents of results and discussions. Please separate them.
- The resolution and performance of figures are not clear.
- The authors give more discussion related to the results of this study.
- There is a logical contradiction in the discussion.
R: The authors acknowledged the reviewer`s comment on manuscript. Author guidelines were checked and required corrections were performed. Statistical analysis of life-table parameters was checked and necessary modifications were added. The results section was modified, please see the section and results and discussion sections were separated. Figures were changed to color version and high-resolution formats were added to the manuscript. Discussion was reworded to reflect the results of the current study. Some of the points were addressed in the discussion section to provide clarity. Please see the section.
Minor
Line 6: Ferdinand Pfab7 à Ferdinand Pfab6.
R: Corrected.
Please add “Simple Summary”.
R: Simple summary was added to the manuscript.
Line 41: Halyomorpha halys à H. halys.
R: The authors prefer to start a sentence with the full wording of the insect and consistently use the abbreviation when the insect is described within the remainder of the sentence., when starting the sentence scientific names should be used as it is not in abbreviation form.
Line 59-62: Please give references.
R: References were cited at the end of the sentence. Please see LN 70-73.
Line 62-63: This sentence is redundant.
R: The sentence was removed
Line 138: There are no results of development under extreme temperatures.
R: The wording was changed to ‘Survival at temperatures below developmental thresholds’, please see the new title 2.2
Line 163: Please check other equation.
R: This is an unclear question, which equation should be checked? The authors checked all equations and they are correct.
Line 166-167: There is no results of the nonlinear estimation model to the intrinsic rate of population increase.
R: This sentenced was deleted, we did not estimate these.
Line 174-175: The authors explain why they choose Gompertz function.
R: Gompertz function adequately describes the collected mortality data of H. halys for both datasets
Line 178: What is “0.235” in the function?
R: It is a constant that is added to the function based on the statistical fit of the curve to describe the collected data
Line 184: The authors explain why they choose Cauchy function.
R: Cauchy function adequately describes the collected distribution of the fecundity data of H. halys for both datasets.
Line 196-197: How the authors define the definition of highest and lowest temperatures in here?
R: These are the trialed temperatures; wording was included to better define these temperatures. Please see LN. 194-195.
Line 197-198: “Adult life span” and “adult to mortality” (in table 1) are the same concept or not?
R: Yes, correct. We meant adult to mortality. The required correction was made on the manuscript. Please see LN 195-197.
Table 1: Why the authors add “Egg to adult” data?
R: These data are essential for estimation of the mean generation time (T). It is data that can be used by scientists for modeling, and provide a summary of these available values for the immature life stages.
Table 1: The column of EFD in New Jersey at 17°C is 100. Is it correct?
R: There is no data on egg laying at 17 C, thanks for pointing this out. That datapoint was removed.
Line 200-202: This sentence is redundant.
R: The sentence was reworded. Please see LN 198-201.
Line 218: at 25 and 30 à at 25 and 22?
R: Thanks for the reviewer`s comment. The number was corrected. Please see LN 219-220.
Line 219-221: This sentence is redundant.
R: Sentence was reworded. Please see LN 220-222.
Line 223: Would you check the contents of table 1?
R: the contents was checked, and an incorrect value was removed and cite in the text was removed.
Figure 2. Color dots are better to read the data.
R: The data was changed to color as requested.
Line 234: How the authors define the definition of highest and lowest temperatures in here?
R: Wording was added for clarity. Please see LN 236-238.
Line 248: The highest EFD values were recorded at 27 and 30 à The highest EFD value was recorded at 30.
R: The sentence was changed as suggested. Please see LN 252-253.
Figure 3. Color dots are better to read the data.
R: The authors changed the dots to color.
Line 259-261: Please add the results of New Jersey population.
R: The immature data for BMSB was only determined in New Jersey, this is clarified in the new text. LN 264-266.
Figure 4: Color dots and lines are better to read the data.
R: Changed as suggested.
Figure 4B. There are two nonlinear curves. Which one is correct?
R: This is correct, there are two curves, the first was fitted to (Fig. 4 A) data collected from Oregon, and the second (Fig. 4b) for data collected from New Jersey. We cannot assume that either dataset or fitting is ‘correct’. These are simply datasets collected and reported, as well as fitted by the appropriate curves. This data provides a comparative spectrum of survival rates based on slightly varying methodologies. We believe that this data is valuable, specifically in this regard, because it shows insect response to these conditions. In addition, the straight line represents the estimated values while dotted line is for proportion alive. I might look like straight, but it is actually multiple data points close each other that it looks like straight line.
Line 268-273: Please make a table for readers.
R: Table 2 was added for model fit of oviposition.
Line 269: F à F, p à P.
R: Changes were made as suggested. LN 274-275.
Line 270: R2 à R2, F à F, p à P.
R: Changes were made as suggested. LN 274-275.
Line 276-277: What are the parameters (l and s)?
R: These are constants selected by the statistical software to fit the distribution to the collected data.
Line 279-282: The contents are estimated EFD?
R: Yes, statement was reworded. Please see LN. 288.
Figure 5: Color dots and lines are better to read the data.
Figure 6: Color dots and lines are better to read the data.
R: Figure 5 &6 were changed to color dots and lines as suggested.
Line 300-302: This sentence does not relate to the results of this study.
R: Paragraph was restructured. Please see LN.337-348.
Line 303-304: Which population did you compare with other pentatomids?
R: Both the Oregon and New Jersey populations and other pentatomids were removed from the text. Please see LN 337-348.
Line 307-308: The authors discuss about this sentence.
R: Additional discussion was added. Please see LN 342-348.
Line 310-321: The authors give discussion focusing on the results of this study.
R: The section was changed please see the LN 342-348.
Line 322-329: Please give quantitative explanation the difference or similarity between two populations.
R: Section was changed please see LN 349-357.
Line 331-344: Why the authors add this part?
R: The winter DD survival was a part of the current work and the authors were providing a discussion for that section. Clarification was added in the text. Please see LN 369-385.
Line 345-366: The authors explain this part clearly and logically.
R: The comment is appreciated.
Line 367-377: This part move to the results.
R: The text was moved as suggested. LN 305-313.
Table 2: Please reconstruct the table 2 for readers.
R: The authors reconstruct the Table 3 (previously Table 2); however, the comment was not clear enough to understand what reviewer was pointing out. Please see Table 3.
Line 368-370: Please rewrite this sentence.
R: The sentence was reworded. Please see LN 306-307.
Line 375: “32” à “30”?
R: 32 was changed to 30 as suggested. Please see LN 312.
Line 389-391: Please give some information about these part in the results.
R: The paragraph was moved to the results section. LN 323-325.
Line 397-399: There is no contents of this part in the results and discussion.
R: There is please see the section 3.2.
There is no conclusion part.
R: The conclusion part was added to the manuscript. Please see the section.
References: The authors follow the guideline of “Journal Insects”.
R: I used Zotero for References and I downloaded the style from Insects website from Author`s guideline section. I double checked; however, I cannot change the style for that reason the reference was not correct in the previous version. I manually changed the references and inform the editors of Insects for the issue.

Reviewer 3 Report
Review: Mermer et al., Temperature-dependent life table parameters of BMSB
Introduction
L53-55. Grammar: “A previous report…” or “Previous reports…”
L55-56. Grammar: mixed pronouns (singular/plural)
L63-64. There are plenty more citations that can be used to support this statement
L66-68. This is not a sentence
L76-78. This isn’t a real sentence, and the point of this sentence is not clear. L69-70 already says some of this.
L76-84. This entire paragraph needs rephrasing
L142. At a temperature below? At temperatures below?
L216-218. This needs rephrasing. Fecundity can’t “progressively” decrease at both higher and lower temperatures. Suffice to say that it was maximal at 25 °C.
Methods
Oregon colony maintained at 22 °C, 16:8 L:D; then tested at 7 temperatures
NJ colony reared at 25 °C, 16:8 L:D; hen tested at the different temperatures
Why were the colonies not reared in the same conditions? This is likely to have pre-conditioned the insects to different temperatures
Diets were also different, and diet affects longevity, survival, and reproduction (including fecundity) of many insect pests. It therefore seems from the methodology that the results are not going to be comparable.
There is nothing in the methodology about Swiss colonies studies, yet the results of these studies are presented alongside the NJ results from this study in Table 1. See below for additional comment on this.
Results
Table 1 only seems to have data at 22 °C from NJ – all other data here appear to be from Nielsen et al. 2008 or Haye et al. 2014. Why are no other results from the NJ experiments provided?
Because Table 1 only includes NJ data at 22 °C, it is not immediately clear what the data in Figure 1c are from: is this NJ data, or is this a plot from Table 1 data?
Headers here for Oregon & NJ are redundant, as data for both places are provided in Table 1 & Fig.1
L258-262. Information provided inconsistently for Oregon & NJ: lots of detail for Oregon & none for NJ. These lines could just be removed entirely.
Fig.4 is a bit confusing. The fitted line is to the points, which is total H. halys lifetime in degree-days (i.e., egg to adult), whereas the legend states it is to adult degree-days only.
L268-271. “…regression…in Oregon population…in the New Jersey population”. Which population are we talking about here? I eventually worked it out, but this needs re-writing as it really does not make sense as it is. Use 2 sentences? Also, why write the equations in the text rather than put them on the figure? The latter would be much simpler and would greatly enhance readability & clarity. It would also make it easier to compare the 2 equations.
L275-282. As above – needs to be clearer. This is badly presented.
L286. What is a daily winter survival curve? Rephrase to say the Gompertz equation fits the data well. And again, equation etc. on the figure rather than just giving parameters in the text.
Discussion
L300-303. How are life table studies on other species at 26 °C comparable or even relevant to your study, which did not use 26 °C?
L304. “abd” should be “and”
L309. “Longevity at this temperature…” – are you still talking about 25 °C?
L300-309. This paragraph needs re-writing. It is very unclear, with information at different temperatures for different species being provided in different places. E.g., at 25 °C, longevity of other pentatomids ranges from 31-41 days, whereas H. halys was found to live for 48 days in a previous study and 29 days in your study.
L310-320. This entire paragraph also needs rewriting. Simply providing this information as new data is not how a discussion should read. You are simply providing new data (other studies on other species) with lots of numbers, but you are not really putting it into any context for H. halys. I’m not sure if the detail (xx.x days) needs to be entirely omitted, with just summaries provided, or how to tackle this.
L324-327. This sentence does not make sense. You start off talking about extreme low and high temperatures, then only talk about the upper bound. Also, what are “survival and reproductive regions”?
L327-330. Does this accord with other data and/or other models?
L333. “..similar to other recordings from H. halys [31].” Perhaps ”…similar to other findings for this species [31].”?
L333-335. Which trialled temperatures? Another study? If so, provide the reference.
L335-337. Is the point of this to state that survival was higher (73-88%) in Virginia [32] than in somewhere else (61%) over a 50-day period [no reference]?
L350-352. I go back to my comment at the beginning: your lab colonies were kept at different temperatures and fed different foods, both of which are known to affect longevity and reproduction. And yet you go on to discuss that you don’t understand why you got different results.
I feel that this is not clearly presented, and I have issues with the methodology. The language needs to be improved; presentation of results is not clear. This reads more like a draft than a publishable paper. I have made numerous comments in the attached document: these need to be addressed.
There is a major flaw in the experimental design, with the colonies being maintained at different temperatures and provided with different foods: both of these factors are known to affect longevity and reproduction/fecundity of other insect species. It is therefore not surprising that the results from the 2 locations are different, yet methodological differences are not even considered in the discussion. I get the feeling that this has not really been thought through correctly.
Author Response
Reviewer#3
Introduction
L53-55. Grammar: “A previous report…” or “Previous reports…”
R: A previous report. Please see LN 64.
L55-56. Grammar: mixed pronouns (singular/plural)
R: Nymphs are… Please see LN. 66.
L63-64. There are plenty more citations that can be used to support this statement
R: Citations were added at the end of the statement. Please see LN 73-74.
L66-68. This is not a sentence
R: The statement was reworded for clarity. Please see LN. 75-78.
L76-78. This isn’t a real sentence, and the point of this sentence is not clear. L69-70 already says some of this.
R: The authors would appreciate the comment of the reviewer. The sentence was reworded and the repeated section was removed from the text. LN. 81-88.
L76-84. This entire paragraph needs rephrasing
R: The authors acknowledge the comment and the paragraph was reworded as suggested. LN. 81-88
L142. At a temperature below? At temperatures below?
R: The statement was reworded. Please see LN 139-140.
L216-218. This needs rephrasing. Fecundity can’t “progressively” decrease at both higher and lower temperatures. Suffice to say that it was maximal at 25 °C.
R: The authors thank the reviewer. The statement was checked and we removed “progressively” from the sentence. Please see the LN 218-219.
Methods
Oregon colony maintained at 22 °C, 16:8 L:D; then tested at 7 temperatures
NJ colony reared at 25 °C, 16:8 L:D; then tested at the different temperatures
Why were the colonies not reared in the same conditions? This is likely to have pre-conditioned the insects to different temperatures
R: Although the rearing conditions in NJ and OR were different i.e., 22 C in Oregon and 25 C in NJ, this may have resulted in some level of preconditioning. The data presented here should therefore consider these differences.
Diets were also different, and diet affects longevity, survival, and reproduction (including fecundity) of many insect pests. It therefore seems from the methodology that the results are not going to be comparable.
R: The authors acknowledge the comment and address the comment in the discussion section. Please see LN.395-399.
There is nothing in the methodology about Swiss colonies studies, yet the results of these studies are presented alongside the NJ results from this study in Table 1. See below for additional comment on this.
R: The author would like to thank the reviewer for pointing out the misplacing. We initially put the values for Egg to Adult and it was a mistake placing the citation by the temperature rather than values on Egg to Adult section. The remaining numbers are from the New Jersey data itself. Please see corrected version of Table 1.
Results
Table 1 only seems to have data at 22 °C from NJ – all other data here appear to be from Nielsen et al. 2008 or Haye et al. 2014. Why are no other results from the NJ experiments provided. The data was presented to provide reference of currently available data on this insect.
Because Table 1 only includes NJ data at 22 °C, it is not immediately clear what the data in Figure 1c are from: is this NJ data, or is this a plot from Table 1 data?
Headers here for Oregon & NJ are redundant, as data for both places are provided in Table 1 & Fig.1
R: The authors would like to keep the headers for the readers.
L258-262. Information provided inconsistently for Oregon & NJ: lots of detail for Oregon & none for NJ. These lines could just be removed entirely.
R: The paragraph was reworded for the clarity. Please see LN. 262-267.
Fig.4 is a bit confusing. The fitted line is to the points, which is total H. halys lifetime in degree-days (i.e., egg to adult), whereas the legend states it is to adult degree-days only.
R: Thanks for pointing out, the legend was fixed. Please see Fig. 4.
L268-271. “…regression…in Oregon population…in the New Jersey population”. Which population are we talking about here? I eventually worked it out, but this needs re-writing as it really does not make sense as it is. Use 2 sentences? Also, why write the equations in the text rather than put them on the figure? The latter would be much simpler and would greatly enhance readability & clarity. It would also make it easier to compare the 2 equations.
R: We acknowledge the reviewer`s comment and we added Table 2 for clarification. Please see the section.
L275-282. As above – needs to be clearer. This is badly presented.
R: The statement was reworded for clarity. Please see LN. 283-291.
L286. What is a daily winter survival curve? Rephrase to say the Gompertz equation fits the data well. And again, equation etc. on the figure rather than just giving parameters in the text.
R: DD winter survival curve
Discussion
L300-303. How are life table studies on other species at 26 °C comparable or even relevant to your study, which did not use 26 °C?
R: The whole paragraph was rewritten. Please see LN. 337-347.
L304. “abd” should be “and”
R: the sentence was removed from the text. LN 337-339.
L309. “Longevity at this temperature…” – are you still talking about 25 °C?
R: Yes, the sentence was reworded. Please see LN 337-338.
L300-309. This paragraph needs re-writing. It is very unclear, with information at different temperatures for different species being provided in different places. E.g., at 25 °C, longevity of other pentatomids ranges from 31-41 days, whereas H. halys was found to live for 48 days in a previous study and 29 days in your study.
R: The statement was added to the paragraph for clarity. Please see 337-347.
L310-320. This entire paragraph also needs rewriting. Simply providing this information as new data is not how a discussion should read. You are simply providing new data (other studies on other species) with lots of numbers, but you are not really putting it into any context for H. halys. I’m not sure if the detail (xx.x days) needs to be entirely omitted, with just summaries provided, or how to tackle this.
R: The authors acknowledge the reviewer`s comment. The entire paragraph was revised and information about the other species was removed from the text. Please see the section LN. 337-347.
L324-327. This sentence does not make sense. You start off talking about extreme low and high temperatures, then only talk about the upper bound. Also, what are “survival and reproductive regions”?
R: Statements were reworded for clarity. Please see LN 351-355.
L327-330. Does this accord with other data and/or other models?
R: Yes, relevant reports and models were cited. Please see LB 355-357.
L333. “..similar to other recordings from H. halys [31].” Perhaps ”…similar to other findings for this species [31].”?
R: The sentence was reworded for clarity, please see LN 369-371.
L333-335. Which trialled temperatures? Another study? If so, provide the reference.
R: Reference was added to the sentence, [28], Lowenstein and Walton, 2018. LN. 375.
L335-337. Is the point of this to state that survival was higher (73-88%) in Virginia [32] than in somewhere else (61%) over a 50-day period [no reference]?
R: The sentence was reworded for clarification. Please see the LN. 375-378.
L350-352. I go back to my comment at the beginning: your lab colonies were kept at different temperatures and fed different foods, both of which are known to affect longevity and reproduction. And yet you go on to discuss that you don’t understand why you got different results.
R: The sentence was removed and an additional statement was added to the text. Please see LN. 388-406.
I feel that this is not clearly presented, and I have issues with the methodology. The language needs to be improved; the presentation of results is not clear. This reads more like a draft than a publishable paper. I have made numerous comments in the attached document: these need to be addressed.
There is a major flaw in the experimental design, with the colonies being maintained at different temperatures and provided with different foods: both of these factors are known to affect longevity and reproduction/fecundity of other insect species. It is therefore not surprising that the results from the 2 locations are different, yet methodological differences are not even considered in the discussion. I get the feeling that this has not really been thought through correctly.
R: The authors included datasets from two populations reared under different conditions. Data was collected and reported based on materials. Authors assert that these data provide a more comprehensive dataset that can help readers see how different experimental conditions result in different responses in H. halys insect populations. The authors believe that this dataset is stronger than a single study, because of these differences. Similar analytical methods were employed, allowing for the creation of comparable life-table parameters. Additional wording was added in the discussion, taking these differences into consideration. Thanks for the comment, we included it, and believe that this perspective provides a clearer rationale for presenting this data in a single manuscript.

Round 2
Reviewer 2 Report
This manuscript provided additional estimates of daily mortality, survival, and reproductive rates, age-specific survival and maternity, and survival at temperature below development threshold of H. halys in two temperate North American.
“In populations having both sexes, data pertaining exclusively to the females will never suffice to represent the true characteristics of the entire population. Application of female age-specific life tables to two-sex populations will consistently result in errors in data analysis, interpretation, prediction and decision making.” (Chi et al., 2020).
Chi H, You M, Atlihan R, Smith CL, Kavousi A, Ozgokce MS, Guncan A, Tuan S-J, Fu J-W, Xu Y-Y, Zheng F-Q, Ye B-H, Chu D, Yu Y, Gharekhani G, Saska P, Gotoh T, Schneider MI, Bussaman P, Gokce A, Liu T-X. 2020. Age-stage, two-sex life table: an introduction to theory, data analysis, and application. Entomologia Generalis 40, 103-124.
The experimental processes (section 2.1) (design, supplied food resources and possible mating number of female) were different between two populations. How the authors can compare with the results of two populations?
Line 162
The authors think about above equation.
Line 171: Those results came from New Jersey population. The authors applied the results to Oregon populations to calculate the DD. There is no lower and upper thresholds data of Oregon populations? If so, the authors need more experiments to get more accurate results.
Table 2. Would you check the results of New Jersey model? Is it correct?
Reviewer 3 Report
I have looked briefly at the new version, and I still do not think it merits publication. Whilst the authors have made many changes, I still maintain that their methodology is flawed, and as such, so are their conclusions. The data and results presented do not add anything useable to the literature. The authors claim that this information can be used to build a detailed population dynamics model of BMSB – this is not possible if different equations must be used in different locations.